# Fusarium Yellows of Ginger (*Zingiber officinale* Roscoe) Caused by *Fusarium oxysporum* f. sp. *zingiberi* Is Associated with Cultivar-Specific Expression of Defense-Responsive Genes

**DOI:** 10.3390/pathogens12010141

**Published:** 2023-01-14

**Authors:** Duraisamy Prasath, Andrea Matthews, Wayne T. O’Neill, Elizabeth A. B. Aitken, Andrew Chen

**Affiliations:** 1School of Agriculture and Food Science, The University of Queensland, St. Lucia, QLD 4067, Australia; 2ICAR—Indian Institute of Spices Research, Kozhikode 673012, India; 3EcoSciences Precinct, Department of Agriculture and Fisheries, Dutton Park, Brisbane, QLD 4102, Australia

**Keywords:** *Fusarium oxysporum*, ginger, Fusarium yellows, plant immunity, gene expression, resistance genes, defense response, host–pathogen interaction

## Abstract

Ginger (*Zingiber officinale* Roscoe) is an important horticultural crop, valued for its culinary and medicinal properties. Fusarium yellows of ginger, caused by *Fusarium oxysporum* f. sp. *zingiberi* (*Foz*), is a devastating disease that has significantly reduced the quality and crop yield of ginger worldwide. The compatible interaction between ginger and *Foz* leading to susceptibility is dissected here. The pathogenicity of two *Foz* isolates on ginger was confirmed by their ability to colonise ginger and in turn induce both internal and external plant symptoms typical of Fusarium yellows. To shed light on *Foz* susceptibility at the molecular level, a set of defense-responsive genes was analysed for expression in the roots of ginger cultivars challenged with *Foz*. These include nucleotide-binding site (NBS) type of resistant (*R*) genes with a functional role in pathogen recognition, transcription factors associated with systemic acquired resistance, and enzymes involved in terpenoid biosynthesis and cell wall modifications. Among three *R* genes, the transcripts of *ZoNBS1* and *ZoNBS3* were rapidly induced by *Foz* at the onset of infection, and the expression magnitude was cultivar-dependent. These expression characteristics extend to the other genes. This study is the first step in understanding the mechanisms of compatible host–pathogen interactions in ginger.

## 1. Introduction

Ginger (*Zingiber officinale* Rosc.) is an economically important horticultural crop that is consumed as a spice and highly valued for its medicinal properties. It has a long history of medicinal use dating back 2500 years in China and India for conditions such as headaches, nausea, rheumatism, and colds [1,2]. Ginger contains various therapeutic compounds with antiemetic, antiulcer, anti-inflammatory, antioxidant, antiplatelet, and anticancer activities [3,4,5,6,7].

Ginger is believed to have originated in Southeast Asia. However, its true centre of origin remains uncertain as it is not found growing in the wild [8,9,10,11]. The long history of ginger cultivation, especially in China and India, has resulted in many cultivars and land races [12]. The cultivars grown for commercial production are essentially infertile. Seed set is rarely observed [10,13]. Ginger is, therefore, clonally propagated like banana, and the lack of viable seeds makes it challenging to produce hybrid progeny in breeding programs. In turn, this makes breeding for disease resistance and environmental adaptability as difficult as it is urgent [14,15,16,17]. Low genetic variability in ginger crops is of particular concern in the Australian context [18].

Australia produces, on average, close to seven thousand tonnes of ginger annually, with a production value of AUD 56 million (AUS dollar) in 2021 [19]. While this volume is expected to increase, it only represents about 1% of worldwide ginger production [18,20]. The Australian ginger industry started in the 1920s with the importation of rhizomes from Cochin, India. This cultivar became known as ‘Queensland’ [21], and it is still the preferred cultivar for processing into confectionary [11,22,23]. The second dominant ginger cultivar grown in Australia is ‘Canton’, also known as ‘Jumbo’. This cultivar also has large, uniform rhizomes but is primarily used in the fresh market [22,24]. It was imported into Australia in the early 1970s to increase the available ginger cultivars [11]. The internal flesh is generally pale yellow, but more mature portions of the rhizome may have a blueish tinge from anthocyanin pigmentation. A third, minor cultivar in Australia is ‘Jamaican’. It tends to be relatively low yielding, and the flesh of mature rhizomes develops an intense blue colour from anthocyanin pigmentation [24]. This cultivar was also imported into Australia for assessment in the early 1970s. The ‘Jamaican’ rhizomes were found to have distinct oil composition and pungency and aroma qualities but lacked the citral aroma found in both ‘Queensland’ and ‘Canton’ rhizomes [25]. Due to its unique properties, the ‘Jamaican’ cultivar has potential in cosmetic industries [25]. Although other ginger cultivars and related species are not cultivated at a commercial scale, they represent a significant component of the germplasm, which could prove a valuable resource for future breeding programs. However, with currently only two dominant commercial cultivars, the Australian ginger industry may be vulnerable to changing global climate conditions and biotic stresses.

Ginger is susceptible to a wide range of biotic and abiotic stresses that adversely affect yield, particularly bacterial and fungal diseases [26,27]. One of the most significant fungal pathogens of ginger is *Fusarium oxysporum* f. sp. *zingiberi* (*Foz*), the causal agent of the disease called Fusarium yellows. Fusarium yellows has been a problem in Australia since the 1930s and is widespread in Asia (India and China), North America (Hawaii), and Oceania (Australia, Hawaii, and Fiji) [9,27,28,29,30,31]. This disease is initially characterised by bright yellowing of the leaves of the affected tiller, while neighbouring tillers may appear healthy. The yellow leaves then dry out, and the disease progresses to the other tillers and leaves, causing the whole above-ground portion of the plant to wither and die. When lifted, the rhizomes are brown with internal rot of the outer layers and brown discolouration in the xylem regions. Some pieces or parts of rhizomes may appear unaffected.

*Fusarium oxysporum* belongs to the phylum Ascomycota; however, the production of ascospores has never been observed in this species [32,33]. *Foz*, the strain of *F. oxysporum* that is pathogenic on ginger, produces asexual conidia and survival spores called chlamydospores [34,35]. ‘Queensland’ and ‘Canton’ ginger cultivars are susceptible to *Foz,* and, under weather conditions conducive to disease, over 70% of the ginger rhizomes may be affected and consequently become unmarketable [36]. Susceptibility to *Foz* is a problem for Australian ginger growers as, despite extensive evaluation, no host resistance to *Foz* has been found in Australia. Understanding the molecular mechanism of *Foz* infection in the three major Australian ginger genotypes is important to evolve suitable management strategies or to develop resistant cultivars.

Molecular work aimed at deciphering plant–pathogen interactions has generally produced valuable insights into genes and pathways that may be important to design strategies for enhancing disease resistance in plants [37]. However, ginger does not yet have a genome reference, and genetic resources are not readily accessible to the research community. This has significantly hampered efforts to explore the pathosystems in this nonmodel crop.

Plant response to biotic and abiotic stress involves reprogramming of the transcriptome. A significant portion of the genes underlying transcriptomic changes appear to be commonly shared amongst plant responses to different environmental stimuli [38]. This suggests the convergence of stress-responsive genes from different pathways. A comparative transcriptome analysis revealed important genes in defense pathways of ginger and mango ginger (*Curcuma amada* Roxb) in response to *Ralstonia solanacearum*, commonly known as bacterial wilt [16]. Potential candidate genes were identified in a previous study [39] and were used to explore ginger–*Foz* interactions based on conserved signalling pathways leading to a disease response.

In this study, we examined the epidemiology of *Foz* infection in ginger plants. To dissect the molecular mechanisms of susceptibility, candidate genes were selected based on their function, and differential expression against *R. solanacearum* [39]. Here, we examined the classical resistant (*R*) genes that recognise pathogen-specific effectors, leading to the activation of plant defense signalling pathways [40]; plant enzymes that are part of cell wall modifications, namely xyloglucan endo-transglycosylase/hydrolase (XTH) [41]; enzymes such as plant glutathione-S-transferases (GST) that are involved in the detoxification of metabolites by conjugating with glutathione, hormone transport, and reduction in oxidative stress [42]; transcription factors from WRKY, and APETALA2/ethylene response factor (AP2/ERF) families that have regulatory roles in biotic and abiotic stress tolerance [43]; and genes involved in the biosynthesis of isoprene/terpenes through the mevalonate (MEP) pathway that are differentially regulated between *C. amada* and *Z. officinale* in response to bacterial wilt [39]. Expression profiles for individual candidates were generated from *Z. officinale* during a time course of *Foz* infection. Root-specific expression of candidate genes was compared amongst three Australian ginger cultivars, namely ‘Queensland’, ‘Jamaican’, and ‘Canton’. This study provides a first step towards understanding the molecular mechanisms of compatible host–pathogen interactions in ginger.

## 2. Materials and Methods

### 2.1. Ginger Cultivars

Ginger cvs. ‘Canton’, ‘Jamaican’, and ‘Queensland’ were established from rhizome pieces from the Maroochy Research Facility, Department of Agriculture and Fisheries, Nambour, Queensland, Australia.

### 2.2. Plant Growth

Ginger plants were grown in pasteurised UQ23 mix (70% composted pine bark and 30% coir) in 200 mm pots in a temperature-controlled glasshouse and were arranged in a randomised design with mixed genotypes. Plants were fertilised with an initial application of a teaspoon of a controlled-release fertiliser per pot. The watering regime was three days a week. The maximum and minimum temperatures in the glasshouse during the experiment were 30 °C and 25 °C, respectively, while relative humidity oscillated between 55% and 80%. Once the plants had reached the five-leaf stage and developed a healthy root system (approximately 50 days post germination), the plants were inoculated with *Foz* spore suspension or mock-inoculated with water.

### 2.3. Foz Isolates

Two *Foz* isolates were used in this study. Both strains were isolated from rhizomes of ginger plants with typical Fusarium yellows symptoms, including leaf yellowing, wilting, and a rotten rhizome. *Foz* ‘Goomboorian’ (UQ6790) and *Foz* ‘Eumundi’ (UQ6440) isolates were originally collected from their respective locations, Goomboorian (S26.0383, E152.7855) and Eumundi (S26.4866, E152.9451), in southeast Queensland.

### 2.4. DNA Extraction, PCR, and Phylogenetic Analysis

DNA extraction of the two *Foz* isolates was performed using a microwave method [44]. A 656 bp fragment of the translation elongation factor 1-alpha (TEF-1α) was PCR-amplified using primers (5′-ATGGGTAAGGARGACAAGAC) and (5′-GGARGTACCAGTSATCATGTT) derived from a previous study [45] and Dreamtaq (Thermo Fisher Scientific, Waltham, MA, USA). PCR was performed according to the manufacturer’s instructions and the conditions specified in [45]. A single amplicon was confirmed on a 1% agarose gel, purified using GeneJET PCR purification kit (Thermo Fisher Scientific, Waltham, MA, USA), and Sanger-sequenced (Australian Genome Research Facility, Melbourne, Australia).

TEF-1α sequence accessions representing four *Fusarium oxysporum* complexes were retrieved from previous studies [45,46,47] and from the NCBI nucleotide database “https://www.ncbi.nlm.nih.gov/nucleotide/ (accessed on 12 December 2022)”. Geneious Prime v 2023.0.1 (Biomatter Pty. Ltd., Auckland, New Zealand) was used to perform the subsequent steps. TEF1-α sequences including *Foz* ‘Eumundi’ and *Foz* ‘Goomboorian’ were aligned using MAFFT v 7.490 [48] and then manually edited over two iterations to remove gaps and derive a consensus alignment sequence of 541 bp from 156 accessions. Bayesian inference was performed using MrBayes v 3.2.6 [49]. The settings included the GTR-G-I model of substitution and two independent analyses on four Markov chain Monte Carlo (MCMC) chains for 2,000,000 generations, with a burn-in rate of 25% for every 1000 generations sampled. *F. delphinoides* (NRRL36160) was used as an outgroup to anchor the whole phylogeny. The tree branches were transformed into a cladogram and visualised in Geneious Prime.

### 2.5. Plant Inoculation

The *Foz* strains were grown on potato dextrose agar (PDA) at 28°C for two weeks. A total of 4–5 mycelial agar plugs were used to inoculate 1 L Erlenmeyer flasks containing 0.5 L potato dextrose broth, which were then shaken on a rotating platform at 100 rpm and 26 °C for five days. The cultures were filtered through two layers of sterile Miracloth, and spore concentration was adjusted to 2.0 × 10^6^ condia/mL for inoculation.

For the pathogenicity trial, approximately 20 mL of spore solution was poured over the surface of each pot, avoiding potting mix near the ginger tillers. The soil surface was then topped with a thin layer of potting mix, and the plants were watered lightly. The control plants were mock-inoculated with water. Twelve plants each of ginger cvs. ‘Canton’, ‘Jamaican’, and ‘Queensland’ were grown for this experiment. The treatments of *Foz* ‘Goomboorian’, *Foz* ‘Eumundi’, and water control consisted of four plants each.

For the gene expression assay, a total of 40 plants were maintained in the glasshouse. Plants were inoculated with *Foz* ‘Goomborian’ only using a root-dipping method. Briefly, the plant roots were rinsed with water, blotted dry, and then dipped in a spore suspension of 2.0 × 10^6^ condia/mL for two hours. Plants were then repotted, and the entire root system of each individual plant was harvested at 1, 6, 12, and 24 h post inoculation (hpi) and 2, 6, and 12 days post inoculation (dpi). Samples were snap-frozen in liquid nitrogen and stored at −80 °C. Three plants were used per time point per cultivar. The uninoculated plants were treated with water and then harvested in the same way as inoculated plants.

### 2.6. Pathogenicity Trial

Plants were harvested at five weeks post inoculation. At harvest, they were rated for disease incidence based on the level of internal discolouration and rotting in the rhizomes. The scale used was (1) completely clean, no discolouration; (2) less than 10% discolouration including edge effects or small patches, where not completely clean; (3) discoloured areas greater than 10% and up to 25%; (4) discoloured areas greater than 25% and up to 50%; (5) discoloured areas greater than 50% and up to 75%; and (6) discoloured areas greater than 75% and up to 100%.

Tissue samples were collected from harvested plants to check for the presence of the inoculated pathogen. A total of 24–41 samples were taken from each cultivar/treatment. Samples were taken from the roots, rhizomes, tiller/rhizome joint (base), and tiller at 5 cm above the rhizome. Under aseptic conditions, tissues were surface-sterilised in 2% bleach (1:5 dilution of 12.5% available chlorine from sodium hypochlorite) for approximately one minute, rinsed twice in sterile distilled water, and blotted dry on sterile paper towel. The samples were then aseptically trimmed to expose fresh tissue and then placed on half-strength PDA plates. Plates were incubated at 26 °C for three to seven days. Plates were then assessed for the presence of *Fusarium oxysporum* (*Fo*) such as spores and colonies [50].

### 2.7. RNA Isolation and cDNA Synthesis

Total RNA was extracted from the three ginger genotypes using the Spectrum Plant Total RNA Kit (Sigma-aldrich, St. Louis, MO, USA). RNA integrity was checked on 1% agarose gel, and RNA concentrations were estimated using a NanoDrop ND-1000 spectrophotometer (NanoDrop Technologies, Wilmington, DE, USA). cDNA was then synthesised using 2 µg of total RNA and a high-capacity cDNA reverse transcription kit (Applied Biosystems, Foster City, CA, USA) according to the manufacturer’s instructions.

### 2.8. Selection of Candidate Genes, Primer Design, and Quantitative PCR Analysis

The candidate genes identified by [39] were used for qPCR analysis. The genes were shortlisted for analysis from the ginger transcriptome database gTDB “http://14.139.189.27/GTDB/ (accessed on 12 December 2022)”, maintained by the Bioinformatics Centre, Indian Institute of Spices Research, Kozhikode, which contains the transcriptomes of ginger and mango ginger post inoculation with *R. solanacearum*. The differentially expressed genes were shortlisted from the previous study [39], and 15 defense-related genes were selected for further validation using qPCR.

### 2.9. Primers for Gene Expression Analysis

Target gene primer sequences used for real-time PCR in this study are listed, along with the control gene *Actin* (Table 1).

### 2.10. Quantitative Real-Time PCR

Real-time PCR analysis was performed using PowerUp SYBR Green Master mix kit (Applied Biosystems, USA) on the LightCycler 96 Real-Time PCR system (Rosch, Spich, Germany). The 20 µL reaction mixture contained 10 µL of PowerUp SYBR Green Master mix, 1 µL of each primer (10 mM), 2 µL of a cDNA template (1:5 diluted from 20 μL of cDNA synthesised from 1 µg of RNA), and 6 µL of sterile distilled water. The thermal conditions were as follows: an initial denaturation step at 94 °C for 5 min, followed by 40 cycles of two-step amplification at 94 °C for 15 s and 60 °C for 45 s. *β*-*Actin* served as an internal control [39]. The expression level of the genes of interest was normalised to that of the constitutively expressed *β*-*Actin* gene by subtracting the cycle threshold (CT) value of the gene of interest from the CT value of *β*-*Actin* (∆CT). Fold change of the transcripts was calculated relative to the control (0 hpi) using the 2^−ddCt^ method [51]. Fold differences were transformed using a binary logarithm (log_2_). PCR amplicons were analysed for specificity by constructing a melt curve in the range between 62 °C and 99 °C. The specificity of each primer pair was confirmed by a single-peak melting curve.

Expression levels were calculated using three biological replicates per genotype per time point, the data were analysed using analysis of variance (ANOVA, Villanova, PA, USA) (*p* < 0.05), and the means were separated by Duncan’s multiple range test using SPSS (version 16.0) (Chicago, IL, USA).

## 3. Results

### 3.1. Morphology and Phylogenetic Analysis of Foz

*Foz* ‘Goomboorian’ and *Foz* ‘Eumundi’ colonies generally have abundant aerial and submerged mycelia in PDA (Figure 1A). Colonies appeared white on the upper surface of the plate, while the obverse view appeared pigmented in pale violet (*Foz* ‘Goomboorian’) or dark purple (*Foz* ‘Eumundi’). No other morphological differences were observed between the two isolates. Microconidia are approximately 3.0–5.0 µm wide and 6.0–14.0 µm long, hyaline, single-celled, oval to reniform, and produced abundantly in false heads on short monophialides. Macroconidia are approximately 3.0–5.0 µm wide and 20.0–30.0 µm long, generally 3 septate, hyaline, sometimes slightly curved, and have an apical hook. Chlamydospores are produced in aged cultures.

TEF-1α sequencing produced a 632 bp nucleotide sequence that is 100% identical between *Foz* ‘Eumundi’ (OQ181219), *Foz* ‘Goomboorian’ (OQ181220), and *Foz* (BRIP44986). They are relatively divergent to *Foz* (BRIP39299), sharing three single nucleotide polymorphisms (SNP) at 344 bp (T/G), 348 bp (G/C), and 354 bp (A/C) within the 632 bp consensus sequence. *Foz* (BRIP39299) also has a single nucleotide (T) insertion between 299 and 300 bp. A phylogenetic tree was built using Bayesian inference (Figure 1B). It produced a topology consistent with the previous study [47] in that the four species complexes of Fusarium were separated as unique clades within the tree. The topology of this phylogeny has already been described in [47]. In the FOSC clade, all nonpathogenic and pathogenic *formae speciales* strains are clustered into two major and several minor clades. The four *Foz* isolates, including *Foz* ‘Eumundi’ and *Foz* ‘Goomboorian’, are clustered together in a minor clade with six *F. oxysporum* f. sp. *cubense* (*Foc*) strains (BRIP62957, BRIP62955, BRIP62933, BRIP59170, BRIP58803, and BRIP44611) and three *F. oxysporum* endophytes (UQ6539 and UQ6522) (Figure 1B). The localisation of the ‘Eumundi’ and ‘Goomboorian’ isolates within the *Foz* subclade of the FOSC added support to their *Foz* identity. The polyphyletic distribution of *Foc* in the FOSC has been well documented [46,47], but this is not seen here with the *Foz* isolates, possibly suggesting a narrow genetic base, consistent with the lack of genetic diversity and VCG found in *Foz* so far.

### 3.2. Pathogenicity Trial

At harvest, inoculated plants showed symptoms considered typical for *Foz* infection (Figure 2). The first visible symptom of infection was the appearance of bright yellow leaves on individual tillers. Over the next two to five days, these tillers turned orange and then brown before they collapsed. All three genotypes showed external and internal symptoms when inoculated with *Foz* (Figure 2A–L). Plants typically had symptoms of leaf yellow and tiller collapsing. Both the ‘Queensland’ and the ‘Jamaican’ plants also showed relatively healthy tillers alongside the diseased tillers, a symptom also typical of *Foz* infection (Figure 2A,C,E,G). Internal symptoms of *Foz* infection included browning and loss of structure of the rhizomes. Initially, the discolouration occurs in the cortex, eventually spreading throughout the rhizome. When challenged with *Foz* ‘Goomboorian’, both the ‘Queensland’ and ‘Canton’ displayed browning extending from the cortex into the stele (Figure 2B,J), whereas ‘Jamaican’ had localised discolouration observed in the cortex of the rhizome, but more extensive rotting was also apparent as blackened, shrivelled tissue (Figure 2F). *Foz* ‘Eumundi’ inoculation produced more severe symptoms compared to *Foz* ‘Goomboorian’. Sections through the rhizome of ‘Queensland’, ‘Jamaican’, and ‘Canton’ inoculated with *Foz* ‘Eumundi’ showed advanced rotting and evidence of complete soft tissue collapse, leaving only fibrous material (Figure 2D,H,L). In contrast, the uninoculated plants did not show any external or internal symptoms of *Foz* (Figure 2M–R). Some blue pigmentation was observed in the uninoculated ‘Jamaican’ rhizomes, typical of this cultivar (Figure 2P).

Harvested rhizomes were cut and rated for the degree of discolouration (Figure 3A). In all three cultivars, ‘Queensland’, ‘Jamaican’, and ‘Canton’, *Foz* ‘Eumundi’ produced a higher degree of discolouration than *Foz* ‘Goomboorian’. However, *Foz* ‘Goomboorian’ still caused significant rhizome discolouration and subsequent rotting. These results demonstrate that both *Foz* isolates were pathogenic on all three cultivars. *F. oxysporum*-like (*Fo*-like) colonies were reisolated from all the inoculated plants screened but were not reisolated from any of the uninoculated controls (Figure 3B). They were consistently reisolated from the roots and rhizomes of every inoculated plant tested. The reisolation rate from these two tissue types was, on average, higher in ‘Queensland’ and ‘Canton’ when they were inoculated with *Foz* ‘Goomboorian’ than when these cultivars were inoculated with *Foz* ‘Eumundi’. However, ‘Jamaican’ rhizome showed a higher reisolation rate of *Foz* ‘Eumundi’ than that of *Foz* ‘Goomboorian’.

*Fo*-like strains were reisolated from the tiller base of all three cultivars when challenged with *Foz* ‘Goomboorian’ (Figure 3B). In contrast, when challenged with *Foz* ‘Eumundi’, Fusarium was only reisolated from the tiller base of ‘Jamaican’. Similarly, *Fo*-like strains were reisolated from the aerial part of the tiller, at 5 cm above ground, in all three cultivars challenged with *Foz* ‘Goomboorian’. In contrast, *Fo*-like strains were only reisolated from the aerial part of the tiller in ‘Queensland’. Considering both the disease ratings and the reisolation results, *Foz* ‘Goomboorian’ showed less internal discolouration than *Foz* ‘Eumundi’ (Figure 2A–F), but it was able to colonise the vascular system of ginger more extensively than *Foz* ‘Eumundi’ (Figure 3B).

The qPCR assays were performed using the cDNA of three ginger cultivars prepared from RNA isolated at 1, 6, 12, and 24 hpi and 2, 6, and 12 dpi. Analysis of each amplification product produced a dissociation curve containing a single peak with a narrow melting temperature, indicating that each primer pair amplified a single predominant product. The expression of defense-related genes at the onset of infection is an important factor in restricting the spread of the pathogen.

### 3.3. NBS-LRR Gene Expression

The expression of *ZoNBS1* and *ZoNBS3* was significantly upregulated at the early time points in all three genotypes (Figure 4A). Amongst the three genotypes, the expression of *ZoNBS1* reached the highest magnitude in ‘Queensland’, peaking at 12 hpi, with a 5.8-fold upregulated expression relative to *ZoActin*. To a lesser extent, *ZoNBS1* was up-regulated by a maximum of 3.3-fold relative to *ZoActin* at 1 hpi in ‘Jamaican’ and a maximum of 2.7-fold relative to *ZoActin* at 6 hpi in ‘Canton’ (Figure 4A). After the expression peaks were reached, all three genotypes showed a downregulation of *ZoNBS1* over time, suggesting that *ZoNBS1* upregulation was specific to the early time points and may be important in regulating plant responses at the onset of infection. *ZoNBS2* expression levels were comparable to that of the control throughout the time course, although it was slightly up-regulated at 1 hpi in ‘Canton’ and 6 dpi in both ‘Queensland’ and ‘Jamaican’ (Figure 4A). *ZoNBS3* transcripts were rapidly induced early on in all three genotypes, starting at 6 hpi in ‘Queensland’ and at 1 hpi in both ‘Jamaican’ and ‘Canton’ (Figure 4A). Up-regulation of *ZoNBS3* was the highest in ‘Canton’ and appeared to be maintained at 1 hpi, 12 hpi, 24 hpi, and 2 dpi, suggesting that, unlike *ZoNBS1*, *ZoNBS3* is associated with the attenuation of the ginger defense response over an extended period. ‘Queensland’ and ‘Jamaican’ showed a similar profile with less magnitude of expression in comparison to ‘Canton’. Taken together, all three *ZoNBS* genes showed differential profiles upon *Foz* challenge, and the magnitude of expression appeared to be cultivar-dependent.

### 3.4. Transcription Factor Expression

*ZoAP2* detected contrasting expression profiles in the three genotypes. *ZoAP2* was significantly downregulated across all time points in ‘Queensland’ (Figure 4B). An upregulation was observed in ‘Jamaican’, specifically at 1 hpi and 6 dpi, which are associated with a 2- and 2.7-fold upregulation relative to *ZoActin*, respectively. In ‘Canton’, *ZoAP2* transcript level remained unchanged relative to the control at all time points. Differential regulation of *ZoAP2* was observed amongst the three genotypes. *ZoERF2* was readily induced in all three genotypes with expression peaks observed towards the late time points, at 2 dpi in both ‘Queensland’ and ‘Canton’ and at 6 dpi in ‘Jamaican’. *ZoERF2* expression was upregulated early on in ‘Jamaican’ to three-fold that of *ZoActin* at 1 hpi, whereas it was upregulated later at 6 hpi and 24 hpi in ‘Canton’ and ‘Queensland’, respectively (Figure 4B). This suggests that the timing of *ZoERF2* induction is dependent on the genotype, and its expression tends to peak late during the time course. *ZoWRKY8* showed a significant upregulation by 1.9- and 1.4-fold compared to *ZoActin* at 12 and 24 hpi, respectively, in ‘Queensland’. However, it was consistently downregulated at most time points relative to *ZoActin* in ‘Jamaican’ and ‘Canton’. Overall, the expression profiles of these transcription factors showed a clear differential regulation of these genes amongst the three genotypes.

### 3.5. Terpenoid-Synthesis-Related Gene Expression

The relative expression levels of *ABC* transporter, *HMG-CoA* reductase, and *HMG-CoA* synthase were clearly modulated among three ginger genotypes over the time course (Figure 4C). The expression of *ZoABC* in ‘Queensland’ was a lot higher compared to other genotypes except at 1 hpi, and its maximum expression reached 5.43-fold that of *ZoActin* at 12 dpi. The expression level of *ZoABC* was mostly downregulated throughout the time course in ‘Jamaican’ and ‘Canton’. The *ZoHMGR* expression profiles show that it was upregulated early in ‘Canton’ during the infection by 2.5-fold that of *ZoActin* at 6 hpi, whereas both ‘Queensland’ and ‘Jamaican’ had a maximum expression of 1.8-fold that of *ZoActin* at 2 dpi (Figure 4C). Lastly, *ZoHMGS* was significantly downregulated in ‘Queensland’ throughout the time course. In ‘Jamaican’, *ZoHMGS* expression was induced early on at 1 hpi and then steadily maintained its expression at 1.2–1.5-fold that of *ZoActin* between 2 and 12 dpi (Figure 4C). In ‘Canton’, *ZoHMGS* level was only induced to 1.5–1.6-fold that of *ZoActin* at 24 hpi, 2 dpi, and 6 dpi. Looking at the whole expression profile, *ZoABC* expression was clearly elevated in ‘Queensland’. *ZoHMGR* and *ZoHMGS* showed highest levels of expression at specific time points in ‘Canton’. Transcript abundance of these genes is clearly varied amongst different cultivars.

### 3.6. Cell Wall Defense Related Gene Expression

*ZoBAC* transcripts were significantly upregulated at 24 hpi and 2 dpi in both ‘Jamaican’ and ‘Canton’. The relative expression of *ZoBAC* was the highest in ‘Jamaican’, by two- and three-fold that of *ZoActin* observed at 24 hpi and 2 dpi, respectively (Figure 4D). In ‘Canton’, *ZoBAC* expression was downregulated during the early phase of the infection at 1–12 hpi. In ‘Queensland’, *ZoBAC* transcript levels did not vary much throughout the time course. Nevertheless, a small but still significant peak was detected at 12 dpi. *ZoCalS1* expression levels did not undergo major changes, with only small but significant peaks of expression observed at 6 hpi, 12 hpi, and 2 dpi in the respective ‘Queensland’, ‘Jamaican’, and ‘Canton’ (Figure 4D). *ZoGST* expression was rapidly induced at 6 hpi before reaching a peak of 4.5-fold that of *ZoActin* at 12 hpi in ‘Queensland’. It was downregulated to 0.4-fold that of *ZoActin* at 6 dpi before a slight recovery to 1.3-fold that of *ZoActin* at 12 dpi. Similarly, ‘Canton’ expression also spiked at 12 hpi, with an increase of three-fold that of *ZoActin*. However, unlike ‘Queensland’, its expression level in ‘Canton’ was initially upregulated to two-fold that of *ZoActin* at 1 hpi and was then maintained at similar levels at 2 dpi and 12 dpi. *ZoGST* expression was maintained at a steady level in ‘Jamaican’, with small but significant upregulations at 12 hpi, 2 dpi, and 6 dpi. Overall, the expression patterns of *ZoGST* can be distinguished amongst the three genotypes on the basis of expression magnitude and how fast the gene expression changed over time.

Among the other cell wall defense associated genes assessed, *ZoMLO12* and *ZoHSP* did not show upregulation in any of the time intervals in the ginger genotypes (Figure 4E). In fact, *ZoHSP* was downregulated at every time point in all three genotypes. On the other hand, *ZoXTG* was rapidly induced in ‘Jamaican’ to high transcript levels greater than four-fold that of *ZoActin* at time points 6 hpi, 24 hpi, and 2 dpi. ‘Canton’ showed expression elevated to similar levels at 6 hpi and 6 dpi. In contrast, ‘Queensland’ had much lower levels of *ZoXTG* transcripts at all time points when compared to ‘Canton’ and ‘Jamaican’.

Overall, out of the 15 candidate genes assessed for expression levels in this study, 4 genes, including *ZoNBS1*, *ZoWRKY8*, *ZoABC*, and *ZoGST,* showed significantly elevated expression at one or more time points in the ginger genotype ‘Queensland’ compared to the other two genotypes. In ‘Jamaican’, four genes, namely *ZoAP2*, *ZoERF2*, *ZoBAC,* and *ZoXTG,* showed cultivar-specific upregulation at one or more time points when compared to the other two genotypes. Lastly, *ZoNBS3*, *ZoHMGR*, and *ZoHMGS* were highly upregulated in a cultivar-dependent manner in ‘Canton’ when compared to the rest.

## 4. Discussion

Plant breeders have selected disease-resistant phenotypes in the field to control plant diseases long before the era of molecular biology. Molecular cloning of genetic determinants underlying resistance revealed *R* genes to be a major type of resistance genes that provide resistance to a diverse range of pathogens [52]. Furthermore, signal transduction mechanisms are often commonly shared for the expression of resistance to unrelated pathogens. This is further explored in studies on plant perception of conserved microbial signatures in microbial effectors [53]. This conservation in pathogen recognition and downstream signalling cascades allow *Foz* susceptibility mechanisms to be explored here.

In this study, we specifically selected three *R* genes that were differentially expressed against bacterial wilt in ginger and then examined their expression profiles in response to *Foz*. Amongst the three genes examined, *ZoNBS*1 and *ZoNBS*3 were rapidly induced upon *Foz* infection, and their expression magnitude was clearly cultivar-dependent. R proteins encoded by the *R* genes analysed in this study belong to the nucleotide-binding site and leucine rich repeat (NBS-LRR) type of *R* genes that have been extensively studied in plant–pathogen interactions [54]. In most cases, *R* gene expressions positively regulate disease resistance in a wide range of plant species including rice, sunflower, cucumber, Arabidopsis, and ginger [55,56,57,58,59]. On the other hand, few genes conferring disease susceptibility have been identified. They include *LOV1*, an NBS-LRR type *R* gene that provides disease susceptibility against *Cochliobolus victoriae* in *Arabidopsis thaliana* [60]. This suggests that *ZoNBS*1 and *ZoNBS*3 may condition disease susceptibility and could potentially be used for enhancing *Foz* resistance through overexpression or knock-out.

Transcription factors represent key molecular switches orchestrating the regulation of plant developmental processes in response to a variety of stresses. The AP2/ERF transcription factor superfamily, found only in plants, includes several genes that encode proteins involved in the regulation of disease resistance pathways [61]. Out of the three transcription factor genes examined in this study, the two AP2/ERF family genes, *ZoAP2*, and *ZoERF2,* showed higher expression in ‘Jamaican’, whereas *ZoWRKY8* showed higher expression in ‘Queensland’, when compared to the rest. Expression of *ZoAP2* and *ZoERF2* was readily induced early on at 1 hpi, followed by an upregulation much later at 6 dpi. *ZoERF2* had an additional upregulation at 12 hpi. This suggests that the transcriptional activation of these genes corresponds independently to the onset of infection and, later, the colonisation of roots by the fungus. In wheat, an AP2 homolog, *TaAP2-15*, positively regulated wheat resistance to *Puccinia striiformis* f. sp. *tritici* [62]. *TaAP2-15* expression spiked at the onset of infection (12 hpi) and then again at 48 hpi, corresponding to the activation of signalling at the onset of infection and the colonisation of the host plant by the fungus, respectively. These observations are consistent with the role of AP2/ERF family members in the activation of pathogenesis-related genes leading to enhanced resistance [63,64,65,66]. On the other hand, the expression of *ZoWRKY8* was comparatively higher in ‘Queensland’ than in ‘Jamaican’ and ‘Canton’, with little to no transcript changes observed in the latter two lines. This suggests that *ZoWRKY8* may provide a good target for dissecting resistance/susceptibility in ‘Queensland’. Differential expression of *ZoWRKY8* was previously reported in ginger and mango ginger [39]. In other plant species, WRKY transcription factors have been shown to play important roles in defense responses either as positive or negative regulators [67,68,69]. Recently, a *WRKY8* ortholog was reported to provide enhanced resistance against pathogen infection as well as drought and salt stress tolerance [70].

To overcome biotic stresses, plants have developed robust defenses such as systemic acquired resistance (SAR) [71] and induced systemic resistance (ISR) [72], which recognise signals from pathogens and activate signal transductions leading to the production of antimicrobial proteins or compounds. Amongst these, terpenes and terpene-derived phytoalexin have been shown to be a versatile defense against pathogenic fungi and viruses [73,74,75,76]. They are synthesised through common diphosphate precursors. In the cytosol, the mevalonic acid (MVA) pathway produces precursors for a wide range of molecules including but not limited to sesquiterpenes and triterpenes [77]. Out of the three enzymes in the MVA pathway, acetoacetyl (AcAc)-CoA thiolase (AACT; EC 2.3.1.9) and 3-hydroxy-3-methylglutaryl-CoA (HMG) synthase (EC 2.3.3.10) catalyse sequential steps of converting three acetyl-CoA molecules into one molecule of HMG-CoA, which is converted into MVA by HMG-CoA reductase (HMGR, EC 1.1.1.34) [78]. The expression of *ZoHMGS* and *ZoHMGR* revealed that both genes were upregulated the highest in ‘Canton’ amongst the three genotypes at 6 hpi and 24 hpi, respectively. This is consistent with the increased expression of HMGR and HMGS detected in the leaves and rhizome of ginger and mango ginger against *R. solanacearum* [39].

The pleiotropic drug resistance transporter family, specifically the ATP-binding cassette (ABC) transporter genes, has been shown to play important roles in plant defenses induced by terpenoids in Arabidopsis and tobacco [79,80]. In this study, ‘Queensland’ had a significant upregulation of *ZoABC* transcripts, not seen in the other two cultivars (Figure 4C). This distinct profile of *ZoABC* expression may be further explored to study compatible and incompatible *Foz* interactions in ginger, as evident from a functional analysis of an ABC transporter in Arabidopsis against fungi pathogens including *Fusarium oxysporum* [79].

Plant defenses are multilayered, and the first line of defense often occurs at the cell wall, where a pathogen must penetrate sites that are often composed of cell wall appositions typically associated with papillae, including callose, lignin, cell wall structure proteins such as arabinogalactan, and cell wall polymers including pectin and xyloglucans [81]. Some of these mechanisms including cell wall strengthening, formation of papilla-like structures at penetration sites, and accumulation of different substances within and between cells were identified in pea–*Fusarium oxysporum* f. sp. *pisi* interaction [82].

At the molecular level, six genes were characterised for expression based on their previous characterised functions relating to the cell wall. These include callose synthases (Cals) that are responsible for wound- and pathogen-induced deposition of callose [83,84]; plant glutathione-S-transferases (GST) recognised for their roles in the detoxification of metabolites by conjugating with glutathione, hormone transport, and reduction in oxidative stress [42]; xyloglucan transglycosylase (XTG) involved in the rearrangement of the cellulose/xyloglucan architecture in the cell wall [41]; heat shock proteins (HSPs) providing additional stress tolerance for a range of biotic and abiotic stresses in plants [85], some of which, e.g., HSP90, have been shown to be a chaperone that plays an important role in modulating the structure and stability of R proteins [86]; and *MILDEW RESISTANCE LOCUS O* (*MLO*) *12*, a triple mutant containing *mlo2*/*mlo6*/*mlo12,* which provided enhanced resistance to powdery mildews but elevated susceptibility to *F. oxysporum* in Arabidopsis [87]. Out of these genes characterised, *ZoBAC*, *ZoCals1*, *ZoXTG*, and *ZoGST* appeared to have cultivar-specific upregulation during the infection process in ginger, suggesting potential transgenic or gene editing targets to control resistance/susceptibility in ginger.

Vegetatively propagated crop plants such as ginger, banana, potato, sweet potato, pineapple, citrus, sugarcane, etc., commonly lack genetic diversity and so are particularly susceptible to diseases and pests. In such cases, significant outbreaks of diseases, such as the late blight of potato caused by *Phytophthora infestans* [88,89,90] and Panama disease of banana caused by *Fusarium oxysporum* f. sp. *cubense* [91], have had catastrophic impacts on people and society. In general, when one or two high-yielding cultivars are predominantly used in an industry, it potentially leads to the decimation of an entire crop by disease until a replacement cultivar with appropriate resistance is found.

Genetic diversity is predicted to be reduced over time during evolution in plant species that rely upon vegetative propagation [92]. Due to the practically sterile nature of cultivated ginger, breeding programs are mostly confined to clonal selections [93]. The pathogenicity of microbes on ginger, however, is increased by the host’s low genetic diversity. Consequently, cultivated gingers are susceptible to a range of fungal, oomycete, and bacterial pathogens, including *Pythium* spp., bacterial wilt caused by *Pseudomonas solaracearum*, and Fusarium yellows [94]. So far, ginger’s response to Fusarium yellows has been largely unexplored at both the epidemiological and molecular level. In this study, the nature of this host–pathogen interaction in the first few hours and days of their molecular dance was reported. This step is critically important for the understanding of disease resistance in ginger and supports the long-term goal of developing ginger cultivars with enhanced resistance to *Foz*.

The current understanding of the Australian isolates of *Foz* is that they likely have little genetic diversity. This observation is based on a study by Stirling in 2004 [36]. The author found that, of the 22 *Foz* isolates reisolated from ginger sourced from nine farms, the isolates all belonged to one vegetative compatibility group (VCG). It is generally considered that a VCG represents a clonal, or near clonal, lineage [95,96]. While this may be a relatively small sample, ginger is grown in a small geographical area in Australia, and these farm sites were representative of the local industry [36]. The *Foz* isolates used in the study by [36] were also used by [97] to study the genetic diversity of *Foz*. Randomly amplified polymorphic DNA analysis was used to create a DNA fingerprint of *Foz* [97]. While this research identified some genetic variation, it was concluded that the Australian population of *Foz* was largely homogenous. This is consistent with the low genetic diversity associated with the *Foz* phylogroup in our phylogenetic analysis (Figure 1B). However, the infection process of the two isolates, *Foz* ‘Goomboorian’ and *Foz* ‘Eumundi’, demonstrated that there are pathogenicity-related differences within the *Foz* population. The significance of the variability in Australian *Foz* isolates is yet to be fully determined; however, both *Foz* ‘Goomboorian’ and *Foz* ‘Eumundi’ were pathogenic on the three ginger cultivars ‘Queensland’, ‘Canton’, and ‘Jamaican’.

It was hypothesised that the low level of variability may be due to a single introduction of the pathogen when the original rhizomes were imported [97]. A potential second introduction of Fusarium was documented, but it was thought to have been eradicated [98]. With the importation of new ornamental or commercial ginger planting material and the increase in the importation of fresh rhizomes for domestic consumption, there is an increased risk of new strains of *Foz* becoming established. It is therefore essential that the epidemiology of Fusarium yellows is investigated to enable resistant ginger cultivars to be developed ahead of any new incursion.

Epidemiological studies of *Fusarium oxysporum* pathogens in other crops have found that they colonise through the roots and then the vascular system of the host [99]. In ginger, it is also likely to invade through the cut surfaces of the rhizome seed piece or through wounds caused by nematodes or insects [98]. In carnation wilt, caused by *Fusarium oxysporum* f. sp. *dianthi* (*Fod*), it was found that infection through nonwounded roots resulted in a milder form of the disease than if the roots were wounded before inoculation with *Fod* spores [100]. As ginger is susceptible to damage from a range of plant-parasitic nematodes [101], wounding of the root and rhizomes is likely to be common and significantly contribute to disease development.

*Foz* isolates were reisolated from the tillers of both *Foz* ‘Goomboorian’ and *Foz* ‘Eumundi’ inoculated ginger plants in the present study. This finding reinforces the need to remove crop residues between crop cycles. It is known that *Fusarium oxysporum* f. sp. *cubense* (*Foc*) produced chlamydospores in the pseudostem of the infected banana plant [99]. As chlamydospores are important survival structures, removal of material containing these spores from the field at harvest may be an important way to reduce inoculum for not only the next crop cycle but potentially for crop cycles for the next decade. Cultural practices to reduce the inoculum load are important measures for disease control, especially when resistant cultivars are unavailable.

## 5. Conclusions

The findings of this study revealed that plants, particularly ginger, in general, have evolved common signal transduction mechanisms for providing resistance against different pathogens. Characterisation of genes and their expressions specifically involved in targeting different aspects of resistance mechanisms may lead to new strategies for the control of Fusarium yellows in ginger. For these genes that may have a direct or indirect interaction with *Foz*, genetic manipulation could be used to potentially introduce enhanced resistance in specific ginger cultivars.

## Figures and Tables

**Figure 1 pathogens-12-00141-f001:**
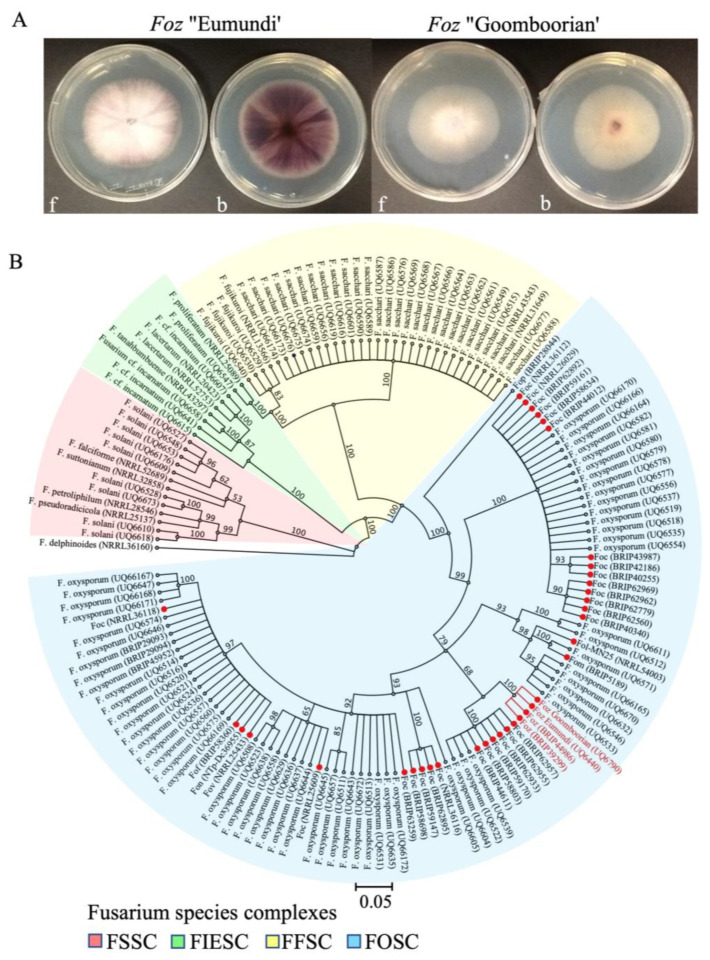
*Foz* ‘Eumundi’ and *Foz* ‘Goomboorian’ morphology and their phylogenetic classifications. (**A**) Colony morphology of the two monoconidial *Foz* isolates, *Foz* ‘Eumundi’ and *Foz* ‘Goomboorian’, grown on full-strength potato dextrose agar for five days. f = front, b = back. (**B**) Bayesian inference phylogeny reconstructed using translation elongation factor 1-alpha (TEF-1α) sequences representing four Fusarium species complexes (FIESC—*Fusarium incarnatum-equiseti* species complex; FFSC—*Fusarium fujikuroi* species complex; FOSC—*Fusarium oxysporum* species complex; FSSC—*Fusarium solani* species complex). Pathogenic *formae speciales* within FOSC are shown with a red circle. Branch labels indicate the posterior probability as a percentage. The *Foz* node within FOSC is highlighted in red, with its members including *Foz* ‘Goomboorian’, *Foz* ‘Eumundi’, *Foz* BRIP44986, and *Foz* BRIP39299. A scale range of 0.05 is indicated below the tree.

**Figure 2 pathogens-12-00141-f002:**
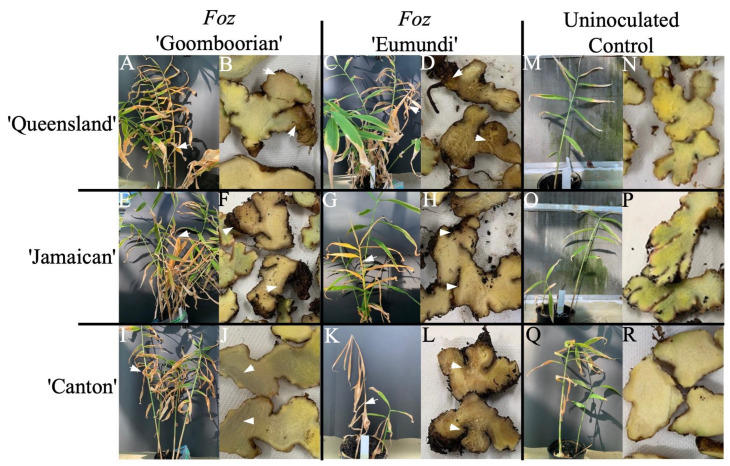
Symptoms of Fusarium yellows in ginger cvs. ‘Queensland’, ‘Jamaican’, and ‘Canton’ when inoculated with either of the two *Foz* isolates, ‘Goomboorian’ (UQ6790) or ‘Eumundi’ (UQ6440). (**A**–**I**) Specific genotype and *Foz* isolate combinations assessed for symptoms on whole plant (left panel) and cut rhizome (right panel). Arrows indicate a typical stunted and yellow symptom on stem and leaves, and brown discolouration associated with the colonisation of the fungus in the vascular tissues of the rhizome. The uninoculated controls were mock-treated with water.

**Figure 3 pathogens-12-00141-f003:**
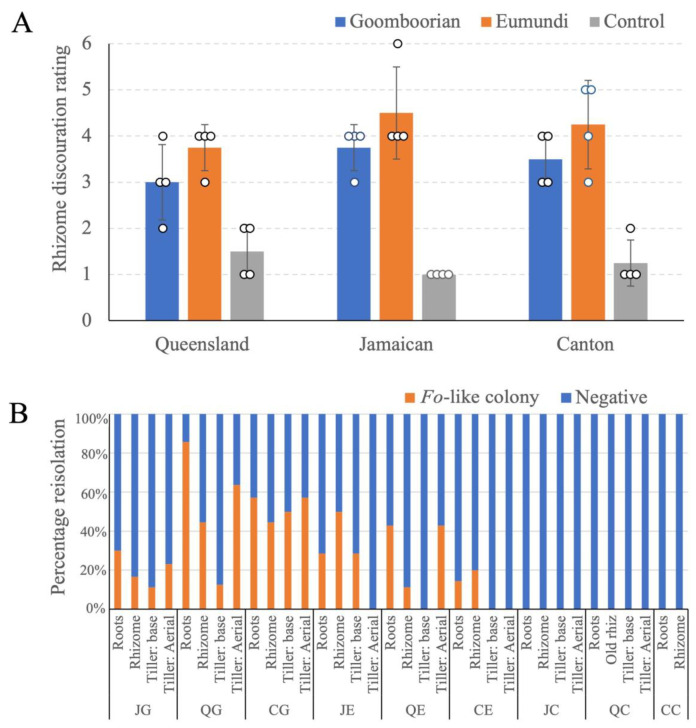
Assessment of symptoms in *Foz* inoculated plants. (**A**) Rhizome discolouration in symptomatic and control plants at full harvest (35 days post inoculation). The number of individual plants assessed per genotype (n) is four. Standard deviations from the means are indicated by error bars. (**B**) Percentage reisolation of *Fo*-like fungi from tissue samples collected at full harvest. First letter indicates the cultivar, J = ‘Jamaican’, Q = ‘Queensland’, and C = ‘Canton’. Second letter indicates the treatment, G = *Foz* ‘Goomboorian’, E = *Foz* ‘Eumundi’, and C = uninoculated control.

**Figure 4 pathogens-12-00141-f004:**
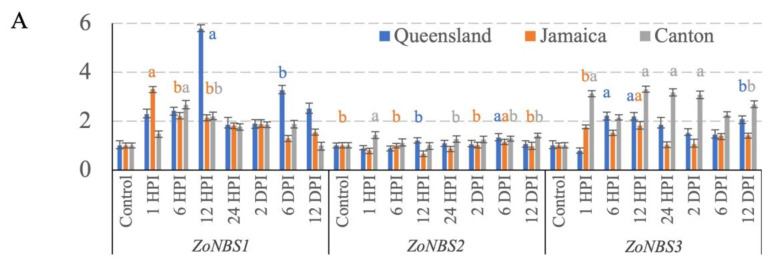
The relative expression levels of target genes against *ZoActin* in *Foz*-infected ginger roots at 1, 6, 12, and 24 h post inoculation (hpi), as well as 2, 6, and 12 days post inoculation (dpi). Gene expression profiles are shown for (**A**) NBS-LRR resistant (*R*) genes *ZoNBS1*, *ZoNBS2*, and *ZoNBS3*. (**B**) Transcription factors, *ZoAP2*, *ZoERF2*, and *ZoWRKY8*. (**C**) Terpenoid-synthesis-related genes, *ZoABC*, *ZoHMGR*, and *ZoHMGS*. Cell-wall-associated genes, (**D**) *ZoBAC*, *ZoCalS1*, and *ZoGST* and (**E**) *ZoMLO12*, *ZoXTG*, and *ZoHSP*. Standard deviation (±SD) is calculated from three biological replicates per genotype per time point. Significant expression differences amongst each genotype were determined by one-way ANOVA. Means were separated by least significant difference (LSD) tests at *p* ≤ 0.05. For each genotype, two expression values with the most significant *p* values are indicated by letters a and b in superscript. Expression profiles are colour-coded according to their genotype. *Z. officinale* cultivar ‘Queensland’ = blue, ‘Jamaican’ = orange, and ‘Canton’ = grey. Colour codes also apply to the statistics performed on each genotype.

**Table 1 pathogens-12-00141-t001:** Primers designed to perform qPCR experiments with the shortlisted 15 candidate genes in ginger and mango ginger [39].

Primer Name	Sequence (5′ to 3′)
*ZoNBS1*-F	TTGGATTCCGGAGGACAAAC
*ZoNBS1*-R	CATCCTAAGAGTGGCCAAGAAG
*ZoNBS2*-F	ACGTGTGGAGCGATGATAAG
*ZoNBS2*-R	ATGTCCCGAGTTGTAACAATGA
*ZoNBS3*-F	GGAAGATGGGCTGGCTTAAT
*ZoNBS3*-R	GCTGGCTTCTCTTTCCTCTAAT
*ZoWRKY8*-F	TGTTTCCATCTCCTACGTCTG
*ZoWRKY8*-R	GTTGGACTTGACCTCATCCT
*ZoAP2*-F	ATGTCAAGGAGCGGCAATAC
*ZoAP2*-R	CATATCCTCGACCGCTTGTTC
*ZoERF2*-F	TCTGAATCCTCCTCGATCAT
*ZoERF2*-R	AGGAGGTTAAGGTCGAAATTG
*ZoABC*-F	TCTGGGACACCTACTGATATTG
*ZoABC*-R	GAATTGCTGATGAAGGGACAG
*ZoHMGR*-F	TGGTCTGTGAAGCAATTATCA
*ZoHMGR*-R	ACCCAGCAAGGTTCTTAATC
*ZoHMGS*-F	TAGATACGGAGCCAAGGATT
*ZoHMGS*-R	GCATAATGTCGACGGTACAT
*ZoMLO12*-F	GTGTATTGCCGCCTTATCTT
*ZoMLO12*-R	CTTTGTAGTTCGATCCCATCTG
*ZoCalS1*-F	GTCCTGAAACCTCTTTCTAGTG
*ZoCalS1*-R	GCAAGGAAGAGAGGCAATAC
*ZoXTG*-F	CAGCAGTTCTACCTCTGGTTC
*ZoXTG*-R	CGTCCCGTCCACATAAATCA
*ZoGST*-F	TGCCGTCTTTGGAGCATAAC
*ZoGST*-R	CGGGATCATCAGGCAGTAATTT
*ZoBAC*-F	GCATCACTGACAGAGAGAAGA
*ZoBAC*-R	GAGCACAAACTGAGGGAGTA
*ZoHSP*-F	AAAGAGGACCAGCTGGAATAC
*ZoHSP*-R	GGTGGTCTTCTCTGTCCATAAA
*ZoActin*-F	TAGGTGCCCAGAGGTTCTATT
*ZoActin*-R	ACCGCTAAGCACCACATTAC

## Data Availability

Not applicable.

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
