# Peer review of "Fusarium Yellows of Ginger (Zingiber officinale Roscoe) Caused by Fusarium oxysporum f. sp. zingiberi Is Associated with Cultivar-Specific Expression of Defense-Responsive Genes"

_pathogens, 2023, doi:10.3390/pathogens12010141_

Round 1

Reviewer 1 Report

The manuscript required some clarification and revision as bellow.

1. Line-5. Remove the full stop from the title.

2. The loss of connctivity in the introduction part is missing. Please give the consize and effective introduction including the importance of zinger, how Foz affected the crop, importance of TF and candidate genes in defecnce response and in last the objective of the study.

3. The authors used three cultivars of zinger but their reaction against Foz is not documented in the manuscript. The contrasting cultivars (HR and HS) will give a clear picture about gene responses. It should be dicussed.

4. line 285-293. It may be deleted. Already given in materials and methods.

5. In fig-4, actin was mentioned with the relative expression. The word “Actin” is not required.

6. The work did not reflacted the candidate genes involved in immunity. Even, the authors also not included the HR genotypes in the study. The title may be revised as “Differential gene expession in zinger cultivars upon Fusarium oxysporum f. sp. zingiberi infection

Author Response

  1. Line-5. Remove the full stop from the title.

Response: corrected.

  1. The loss of connctivity in the introduction part is missing. Please give the consize and effective introduction including the importance of zinger, how Foz affected the crop, importance of TF and candidate genes in defecnce response and in last the objective of the study.

Response: the abstract has been re-written as per reviewer’s instructions.

  1. The authors used three cultivars of zinger but their reaction against Foz is not documented in the manuscript. The contrasting cultivars (HR and HS) will give a clear picture about gene responses. It should be dicussed.

Response: We have now changed the abstract to make it clear that we are exploring a compatible interaction in ginger. There are currently no known resistance sources in ginger cultivars in Australia. We made this clear by making the following statement. Line 86-88:

“Susceptibility to Foz is a problem for Australian ginger growers as, despite extensive evaluation, no host resistance to Foz has been found in Australia.”

  1. line 285-293. It may be deleted. Already given in materials and methods.

Response: deleted the following on line 285-289.

“and relative fold changes with respect to control (0 hpi) plants after inoculation with Foz were calculated. Gene expression analysis of the selected defence-related genes at different time points showed that the genes were expressed early (1-12 hpi), middle (1-2 dpi) and late (6-12 dpi) time periods after pathogen inoculation.”

  1. In fig-4, actin was mentioned with the relative expression. The word “Actin” is not required.

Response: deleted ‘Actin’ from Figure 4.

  1. The work did not reflacted the candidate genes involved in immunity. Even, the authors also not included the HR genotypes in the study. The title may be revised as “Differential gene expession in zinger cultivars upon Fusarium oxysporum f. sp. zingiberi infection

Response: We agree with you. We removed ‘Plant Immunity’ and added ‘defense responsive genes’.

“Fusarium yellows of ginger (Zingiber officinale Roscoe) caused by Fusarium oxysporum f. sp. zingiberi is associated with cultivar-specific expression of defense responsive genes.”

Reviewer 2 Report

The study is a scientifically sound and well-established methodology and findings. However, there are a few major concerns as below. Have done more than 60 edits and plea go through them carefully. 

1.       authors claimed that the isolate is Foz. However,  it is not clear how sure the species is indeed Foz? It is well known that identifying species of Fusarium is really challenging and it is often practiced multilocus phylogeny approach. Authors are encouraged to provide more details of the pathogen identification since molecular mechanisms are seriously depend on the strains and f. sp. at times.

2.       What were the najor differneces between Foz ‘Eumundi and the other strain, Foz ‘Goomboorian? Was there any morphological or molecular level ? or it is just based on isolation ?

3.       Were these 2 isolates belonging to different VCG? At least that should have been mentioned in the methods or anywhere..

4.       Authors have to discuss some cases such as ZoAP2 gene expression. One hour and 6DPI had showed upregulation. Explain what would have been the reason.? Any other similar studies and potential reasons behind such observations.

In addition, authors should follow the journal formatting for example justify the main texts. 

Minor comments are given below.

Line 15                              Remove word count, 199

Line 16                              Ginger (Zingiber officinale Roscoe) … Roscoe should not be in italic throughout the paper

Line  17                             Fusarium yellows is the name of the disease and at this point, it is not necessary to italicize the word, Fusarium. Please maintain throughout the MS and do the changes

Line 17 and 18                 The abstract should re-write. For example, statements like --Ginger is susceptible to fungal diseases, particularly, Fusarium yellows caused by Fusarium oxysporum f. sp. zingiberi (Foz) is quite an argument. If so, what about the other bacteria, viruses and other pathogenic diseases? Authors can present in different ways. (Disease caused by the Fusarium is one of the devastating diseases in Ginger…/Fusarium is one of the prominent diseases of Ginger...)

Lien 19: resistance (R) genes change to resistant (R) genes

Line 29: bacterial should be replaced with bacteria

Lien 30: replace a first step with the first step

Line 32-34                         Fusarium oxysporum 1; Ginger 2; Fusarium yellows 3; plant immunity 4; gene expression 5; resistance genes 6; defense response 7; host-pathogen interaction..........In Keywords, you have used numbers. Please remove those.

Lien 42: originated from replace with originated in

Lien 42-43; change the sentence stricture for easy reading. though, as it is not.. terms disturb the easy reading. or divide the sentence into two.

Ginger is believed to have been originated from Southeast Asia, though, as it is not found growing in the wild, its true centre of origin remains uncertain

Line 46--- viable seed rarely, if ever, observed: rephrase ..with dramatic reduction of seed viability

Lien 46-48: rephrase.. consider revising

One suggestion would be:

Therefore, ginger is clonally propagated like bananas, and the lack of viable seeds makes it challenging to produce hybrid progenies in breeding programs.

Lien 49: is of a

Line 52                              What is meant by---of $50 million AUD in 2021? Is it USD? AUD?

Line 52-53: Remove “This small but growing… section and start sentence with Australian..

Line 55: avoid repeating “this” twice in the same sentence

Lien 68-70: Other ginger cultivars and related species are present in Australia, including native species. Although they are not grown at a commercial level, they represent a source of potential genetic diversity which could prove a valuable resource for future breeding program

Consider combining these sentences or make it reader friendly..

Although other ginger cultivars and related species are not cultivated at a commercial scale, they represent a significant component of the germplasm which could prove a valuable resource for future breeding programs

Line 71                              Replace --- future breeding program.—with---future breeding programs.

Line 84: what s water-conducting regions? Not clear? Is this referring to the xylem?

Line 87                the strain of Fusarium oxysporum that---from use F. oxysporum

Line 92:  is it referring to the global condition or Australian context? Make it clear

Line 115                            better italicize gene names including  'R' genes. Maintain throughout the MS.

Lien 119: by conjugation with glutathione or via conjugation with glutathione or by conjugating with glutathione?

Lien 137: add a space between the number and unit throughout the MS

Lien 159: what was the composition of the potting mix?

Lien 166: individual plants was replace with the individual plant was

Clearly indicate how these plants were maintained ..Watering regime, fertilizer regime.. if any..

Line 171                            Pathogenicity Trial—replace with--Pathogenicity trial

Line 180                            24 - 41 samples---- replace with---24-41 samples

Line 182: how long in bleach solution? Mention is it commercial bleach? What was the concentration of sodium hypochlorite?

Line 190: country for the kit?

Line 197: Karthika et al. in 2018 format as per journal requirement

Lien 202: from the previous study….. now clear what previous study it is? In this study or what was the ref?

Line 213-215: try to use concentrations whenever possible, not in volumes

Line 227: remove technique

Line 232                            Pathogenicity Trial—replace with--Pathogenicity trial

Line 233: Control and treatment plants were harvested at 5 weeks and should be in methods.. try to combine it with the next sentence

Figure 1 needs a pict. of a healthy plant. If control is included in the pictures clearly mention what it was. It would have been better if fig 1 and 2 are combined side by side plant and rhizome for each cultivar. Then control plant and rhizome at last side by side for clarity..

Lien 267 and many places in this para: the use of word recovery sounds like plant recover from the disease.. better use re-isolation or some other term. Same with figure 3 also

Fig 3: C = uninoculated control should it be CC?

Line 285-286: repeating as in methods

Line 358                            Never-the-less----replace with ----Nevertheless,

Line 434: Vegetatively propagated crop plants such as ginger, banana, potato, sweet potato, pineapple, citrus, sugarcane, etc., are of great importance for providing nutrition as well as social and economic security in developing and developed countries

This sounds like other veggies that are having sexual reproduction are not nutritious.. J make it sound

Better to combine 1 and 2nd sentences in discussion in to one..

The discussion needs re written.. start with the most significant finding of the study.. these general statements should be avoided or use later..

Lien 465: In this study,, this study? or by those authors? Not clear

Until the middle of the discussion nothing written about gene expression though it was the major point in paper.. pleas ere write the whole discussion in such a way to bring the key points highlighted..

Line 590-597                    try to avoid using non-quantitative terms lile 'may' . -The findings of the study revealed that plants, particularly ginger, in general, have evolved common signal transduction mechanisms for providing resistance against different pathogens.

Most of the references are not follow the journal guidelines, so all the references should re-check them again. I have summarised some references to get understand.

627                      Reference 5                      Is this reference style correct? Please remove, Chapter 13

Line 630                            Roscoe --should not be in italic. See 15, 24, 30 references too.

Line 632                                           A brief review.-- should be - a brief...... See reference no 17 too.

Line 677                                           Bangladesh j. agric. res. ?? please check it

Line 679                                           First report of Fusarium yellows--- Please check Reference 49 as well.

Line 686                                           Queensland Agricultural Journal? please follow the journal guidelines. Similarly the other references as well. e.g., ref.no 34, 36, 39 and so on.

Figure 4: Y axis can have a single axis name.. 

Author Response

  1. authors claimed that the isolate is Foz. However,it is not clear how sure the species is indeed Foz? It is well known that identifying species of Fusarium is really challenging and it is often practiced multilocus phylogeny approach. Authors are encouraged to provide more details of the pathogen identification since molecular mechanisms are seriously depend on the strains and f. sp. at times.

Response: Based on TE1-alpha sequence of the two Foz isolates, and publicly available sequence data generated from several previous studies, we have constructed a Bayesian inference phylogenetic tree consisting of four distinct Fusarium species complexes, one of them being the Fusarium oxysporum species complex (FOSC). We have demonstrated that our Foz isolates form a subclade with two known Foz isolates, as well as 6 F. oxysporum f. sp. cubense strains, and 3 F. oxysporum endophytes within the FOSC. This analysis provide support to our claim that both ‘Eumundi’ and ‘Goomboorian’ isolates are identified as Foz.

Figure 1. Foz strains grown on PDA, and the phylogenetic tree constructed using Bayesian inference. Line 256-268.

Results corresponding to Figure 1. Line 278-307.

  1. What were the najor differneces between Foz ‘Eumundi and the other strain, Foz ‘Goomboorian? Was there any morphological or molecular level ? or it is just based on isolation ?

Response: Results now include a paragraph describing the morphology of our Foz isolates. Phylogenetic grouping of our isolates within FOSC (Fusarium oxysporum species complex) confirmed their F. oxysporum identity and the formae speciales of both isolates are most likely to be Foz since they formed a sub-clade with 2 other known Foz isolates in the phylogenetic tree.

This result discussed on Line 278-307.

  1. Were these 2 isolates belonging to different VCG? At least that should have been mentioned in the methods or anywhere..

Response: We’ve not determined the VCG of Foz ‘Eumumdi’ and Foz ‘Goomboorian’. The paper by Stirling in 2004 mentioned that a collection of 22 Foz isolates recovered from nine ginger farms in Queensland, Australia all belonged to one VCG. This statement is made in the introduction, line 457-459. So, in order to confirm this, we’d have to perform the VCG test ourselves.

  1. Authors have to discuss some cases such as ZoAP2 gene expression. One hour and 6DPI had showed upregulation. Explain what would have been the reason.? Any other similar studies and potential reasons behind such observations.

Response: Yes, a wheat AP2 homolog, TaAP2-15, appeared to have two expression spikes at 12hpi, and 2dpi, in a compatible interaction against Puccinia striiformis f. sp. tritici (Hawku et al. 2021, doi: 10.3390/ijms22042080).  Similar to our study, the expression level of TaAP2-15 at the late time point (2dpi) was much higher than that at the onset of infection (12hpi). The authors suggest that 2dpi point corresponds to the primary infection via the establishment of fungal structures inside the plant, whereas the signalling events, such as the ROS burst, occurred as early as 12hpi.

We added the following sentences to discuss this observation in the discussion. Line 534-541.

“Expression of ZoAP2 and ZoERF2 was readily induced early on at 1 hpi, followed by an upregulation much later at 6dpi. ZoERF2 has an additional upregulation at 12hpi. This suggests that the transcriptional activation of these genes correspond independently to the onset of infection and later, the colonisation of the roots by the fungus. In wheat, a AP2 homolog, TaAP2-15, positively regulated wheat resistance to Puccinia striiformis f. sp. tritici (Hawku et al. 2021). TaAP2-15 expression spiked at the onset of infection (12 hpi) and then again at 48 hpi, corresponding to the activation of signalling at the onset of infection and the colonisation of the host plant by the fungus, respectively.”

In addition, authors should follow the journal formatting for example justify the main texts.

Minor comments are given below.

Line 15                              Remove word count, 199

Response: removed.

Line 16                              Ginger (Zingiber officinale Roscoe) … Roscoe should not be in italic throughout the paper

Response: corrected.

Line  17                             Fusarium yellows is the name of the disease and at this point, it is not necessary to italicize the word, Fusarium. Please maintain throughout the MS and do the changes

Response: corrected.

Line 17 and 18                 The abstract should re-write. For example, statements like --Ginger is susceptible to fungal diseases, particularly, Fusarium yellows caused by Fusarium oxysporum f. sp. zingiberi (Foz) is quite an argument. If so, what about the other bacteria, viruses and other pathogenic diseases? Authors can present in different ways. (Disease caused by the Fusarium is one of the devastating diseases in Ginger…/Fusarium is one of the prominent diseases of Ginger...)

Response: the abstract has been re-written according to the reviewer’s suggestions, and as per change on line 15-16. “Fusarium yellows of Ginger, caused by Fusarium oxysporum f. sp. zingiberi (Foz), is a devastating disease that has significantly reduced the quality and crop yield of ginger worldwide.”

Lien 19: resistance (R) genes change to resistant (R) genes

Response: corrected on line 23 (original line 19).

Line 29: bacterial should be replaced with bacteria

Response: this word is no longer part of the newly written abstract.

Lien 30: replace a first step with the first step

Response: Corrected.

Line 32-34                         Fusarium oxysporum 1; Ginger 2; Fusarium yellows 3; plant immunity 4; gene expression 5; resistance genes 6; defense response 7; host-pathogen interaction..........In Keywords, you have used numbers. Please remove those.

Response: Corrected.

Lien 42: originated from replace with originated in

Response: corrected.

Lien 42-43; change the sentence stricture for easy reading. though, as it is not.. terms disturb the easy reading. or divide the sentence into two.

Ginger is believed to have been originated from Southeast Asia, though, as it is not found growing in the wild, its true centre of origin remains uncertain

Response: split the sentence into two as per suggestion.

“Ginger is believed to have originated in Southeast Asia. However, its true centre of origin remains uncertain as it is not found growing in the wild [8-11].”

Line 46--- viable seed rarely, if ever, observed: rephrase ..with dramatic reduction of seed viability

Response: We think that it’s not the reduction of seed viability, but rather they rarely set any seeds. We have rephrased these two sentences to make this clearer.

The cultivars grown for commercial production are essentially infertile. Seed set is rarely observed [10,13].”

Lien 46-48: rephrase.. consider revising

One suggestion would be:

Therefore, ginger is clonally propagated like bananas, and the lack of viable seeds makes it challenging to produce hybrid progenies in breeding programs.

Response: made the following correction.

“Ginger is therefore clonally propagated like banana, and the lack of viable seeds makes it challenging to produce hybrid progeny in breeding programs.”

Lien 49: is of a

Response: corrected. (Line 46 current)

Line 52                              What is meant by---of $50 million AUD in 2021? Is it USD? AUD?

Response: added (AUS dollar).

Line 52-53: Remove “This small but growing… section and start sentence with Australian..

Response: started the sentence with “The Australian ginger industry…”

Line 55: avoid repeating “this” twice in the same sentence

Response: Changed the part to ‘it is still the preferred cultivar….’

Lien 68-70: Other ginger cultivars and related species are present in Australia, including native species. Although they are not grown at a commercial level, they represent a source of potential genetic diversity which could prove a valuable resource for future breeding program

Consider combining these sentences or make it reader friendly..

Although other ginger cultivars and related species are not cultivated at a commercial scale, they represent a significant component of the germplasm which could prove a valuable resource for future breeding programs

Response: deleted the original sentences (further above) and added the reviewer’s recommended sentence.

Line 71                              Replace --- future breeding program.—with---future breeding programs.

Response: corrected.

Line 84: what s water-conducting regions? Not clear? Is this referring to the xylem?

Response: changed it to ‘the xylem regions’.

Line 87                the strain of Fusarium oxysporum that---from use F. oxysporum

Response: corrected.

Line 92:  is it referring to the global condition or Australian context? Make it clear

Response: modified the sentence to make it clear that we are talking about host resistance in Australian cultivars. Line 86-88:

“Susceptibility to Foz is a problem for Australian ginger growers as, despite extensive evaluation, no host resistance to Foz has been found in Australia.”

Line 115                            better italicize gene names including  'R' genes. Maintain throughout the MS.

Response: corrected.

Lien 119: by conjugation with glutathione or via conjugation with glutathione or by conjugating with glutathione?

Response: Modified to ‘by conjugating with glutathione’.

Lien 137: add a space between the number and unit throughout the MS

Response: corrected.

Lien 159: what was the composition of the potting mix?

Response: added this information at line 130-131.

“Ginger plants were grown in pasteurised UQ23 mix (70% composted pine bark, 30% coir) in 200mm pots….”

Lien 166: individual plants was replace with the individual plant was

Clearly indicate how these plants were maintained ..Watering regime, fertilizer regime.. if any..

Response: added the following on line 132-133.

“Plants were fertilised with an initial application of a teaspoon of a controlled-release fertiliser per pot. Watering regime was three days a week.”

Line 158-159: added the following.

“a total of 40 plants were maintained in the glasshouse.”

Line 162: individual plants changed to “the individual plant”.

Line 171                            Pathogenicity Trial—replace with--Pathogenicity trial

Response: corrected.

Line 180                            24 - 41 samples---- replace with---24-41 samples

Response: corrected.

Line 182: how long in bleach solution? Mention is it commercial bleach? What was the concentration of sodium hypochlorite?

Response: added the following.

“(1:5 dilution of 12.5% available chlorine from sodium hypochlorite) for approximately 1 minute”

Line 190: country for the kit?

Response: added “AUS”.

Line 197: Karthika et al. in 2018 format as per journal requirement

Response: “The candidate genes identified by [39]”

Lien 202: from the previous study….. now clear what previous study it is? In this study or what was the ref?

Response: added the reference. See below.

“The differentially expressed genes were shortlisted from the previous study [39]….”

Line 213-215: try to use concentrations whenever possible, not in volumes

Response: It would be hard to measure the concentration of the cDNA because it contains all the random hexamers. But there was an error with that statement and was corrected in the following:

 “2 µL of a cDNA template (1:5 diluted from 20uL cDNA synthesized from 1 µg RNA)”

Line 227: remove technique

Response: corrected.

Line 232                            Pathogenicity Trial—replace with--Pathogenicity trial

Response: corrected.

Line 233: Control and treatment plants were harvested at 5 weeks and should be in methods.. try to combine it with the next sentence

Response: removed this sentence, and then added “At harvest”. Line 240.

“At harvest, inoculated plants showed symptoms considered typical for Foz infection.”

Figure 1 needs a pict. of a healthy plant. If control is included in the pictures clearly mention what it was. It would have been better if fig 1 and 2 are combined side by side plant and rhizome for each cultivar. Then control plant and rhizome at last side by side for clarity..

Response: Figure 1 and 2 have been combined as per reviewer’s request. The combined figure is presented as Figure 2. The corresponding results section has been combined and is now on line 306-325.

Lien 267 and many places in this para: the use of word recovery sounds like plant recover from the disease.. better use re-isolation or some other term. Same with figure 3 also

Response: modified appropriately in this paragraph and throughout the manuscript.

Figure 3 has been modified as per reviewer’s suggestion.

Fig 3: C = uninoculated control should it be CC?

Response: Clarification: that is the uninoculated control of ‘Canton’. Uninoculated control (C) of Eumundi is EC, of Goomboorian is GC, of Canton is CC.

Line 285-286: repeating as in methods

Response: this part is deleted.

Line 358                            Never-the-less----replace with ----Nevertheless,

Response: modified.

Line 434: Vegetatively propagated crop plants such as ginger, banana, potato, sweet potato, pineapple, citrus, sugarcane, etc., are of great importance for providing nutrition as well as social and economic security in developing and developed countries

This sounds like other veggies that are having sexual reproduction are not nutritious.. J make it sound

Better to combine 1 and 2nd sentences in discussion in to one..

Response: Agreed. The following bit is deleted.

“are of great importance for providing nutrition as well as social and economic security in developing and developed countries. However, vegetatively reproduced crops”

The discussion needs re written.. start with the most significant finding of the study.. these general statements should be avoided or use later..

Response: Discussion has been restructured. The discussion on the gene expression has been brought forward as it contains the most significant finding of this paper.

Lien 465: In this study,, this study? or by those authors? Not clear

Response: Removed ‘In that paper, it was” and replaced it with “The author”

Until the middle of the discussion nothing written about gene expression though it was the major point in paper.. pleas ere write the whole discussion in such a way to bring the key points highlighted..

Response: as per previous comment – discussion has now been restructured to reflect the significant findings of this study first. The Foz epidemiology now comes after the gene expression sections of the discussion.

Line 590-597                    try to avoid using non-quantitative terms lile 'may' . -The findings of the study revealed that plants, particularly ginger, in general, have evolved common signal transduction mechanisms for providing resistance against different pathogens.

 Response: fixed this as per reviewer’s suggestion.

Most of the references are not follow the journal guidelines, so all the references should re-check them again. I have summarised some references to get understand.

627                      Reference 5                      Is this reference style correct? Please remove, Chapter 13

Response: Fixed

Line 630                            Roscoe --should not be in italic. See 15, 24, 30 references too.

Response: Fixed

Line 632                                           A brief review.-- should be - a brief...... See reference no 17 too.

Response: Fixed

Line 677                                           Bangladesh j. agric. res. ?? please check it

Response: Fixed. Bangladesh j. of agric. res

Line 679                                           First report of Fusarium yellows--- Please check Reference 49 as well.

Response: Fixed

Line 686                                           Queensland Agricultural Journal? please follow the journal guidelines. Similarly the other references as well. e.g., ref.no 34, 36, 39 and so on.

Response: corrected. The correct journal abbreviations have been applied to the entire reference list.

Figure 4: Y axis can have a single axis name..

Response: Fixed.

Additional note to Reviewer 2: Thank you so much for providing such a thorough edit. It has much improved the quality of the paper - we really appreciated your efforts in this regard.

Reviewer 3 Report

None

Author Response

No comments required by the reviewer.